# Understanding the Factors Influencing Junior Doctors’ Career Decision-Making to Address Rural Workforce Issues: Testing a Conceptual Framework

**DOI:** 10.3390/ijerph17020537

**Published:** 2020-01-15

**Authors:** Beatriz Cuesta-Briand, Mathew Coleman, Rebekah Ledingham, Sarah Moore, Helen Wright, David Oldham, Denese Playford

**Affiliations:** 1Rural Clinical School of Western Australia, Faculty of Health and Medical Sciences, University of Western Australia, West Busselton 6280, Australia; beatriz.cuestabriand@rcswa.edu.au (B.C.-B.); mathew.coleman@rcswa.edu.au (M.C.); bek.ledingham@rcswa.edu.au (R.L.); helen.wright@rcswa.edu.au (H.W.); denese.playford@rcswa.edu.au (D.P.); 2Western Australia Country Health Service, Perth 6000, Australia; david.oldham@health.wa.gov.au

**Keywords:** early career, training pathways, postgraduate medical officer

## Abstract

Medical graduates’ early career is known to be disorienting, and career decision-making is influenced by a complex set of factors. There is a strong association between rural background and rural undergraduate training and rural practice, and personal and family factors have been shown to influence workplace location, but the interaction between interest, training availability, and other work-relevant factors has not yet been fully explored. A qualitative study conducted by the Rural Clinical School of Western Australia (RCSWA) and WA Country Health Service (WACHS) explored factors influencing the decision to pursue rural work among junior doctors. Data collection and analysis was iterative. In total, 21 junior doctors were recruited to participate in semi-structured telephone interviews. Two main themes relating to the systems of influence on career decision-making emerged: (1) The importance of place and people, and (2) the broader context. We found that career decision-making among junior doctors is influenced by a complex web of factors operating at different levels. As Australia faces the challenge of developing a sustainable rural health workforce, developing innovative, flexible strategies that are responsive to the individual aspirations of its workforce whilst still meeting its healthcare service delivery needs will provide a way forward.

## 1. Introduction

The provision of adequate healthcare to rural populations is a global issue [1]. In Australia, despite government initiatives implemented in recent years, including increased funding to support medical rural medical training [2], medical workforce shortages and maldistribution between urban and rural locations are ongoing issues [3]. 

The Rural Clinical Schools (RCS) program was implemented in 2000 and mandates that 25% of domestic medical students train rurally for at least one year of their clinical training [3]. Workforce outcomes for the RCSs program have been positive [4,5,6], and a recent study using combined data from multiple RCSs has showed that, independently of student background, RCS students had an increased chance of working rurally five years post-graduation [3]. Despite the program’s success, evidence shows that at best, 50% of rural background RCS alumni end up in rural and regional areas [7]. It is not yet known whether this relatively modest uptake is due to lack of interest or lack of opportunity. 

Medical graduates’ early career is known to be disorienting [8], and career decision-making is influenced by a complex set of factors [9]. There is a strong association between rural background and rural undergraduate training and rural practice [10], and personal and family factors have been shown to influence workplace location [11], but the interaction between interest, training availability, and other work-relevant factors has not yet been fully explored.

Recently, Pfarrwaller and colleagues [12] developed a conceptual framework of medical students’ primary care career choice to explain the various factors influencing final career choices. Although the authors focused on primary care, the model relates well to any specialty and to location practice choices. The model consists of two parts: A central part representing students’ career choice pathways, and an outer part representing the different systems of influence on career decision-making. The central part of the model broadly draws from social cognitive theory as applied to career choice [13], with students entering their training with their personal characteristics and initial interests in primary care, emerging at graduation with a choice for their future career. Students are influenced during their training by outside factors, drawing from ecological theory [14]. The model consists of four nested components representing different systems of influence: The microsystem consists of the settings containing the person; the mesosytem comprises the interrelations among the settings; the exosystem comprises social structures impinging on the settings; and the macrosystem refers to overarching cultural and institutional patterns [12,14].

This study seeks to test Pfarrwaller et al.’s conceptual framework [12] in junior medical graduates’ career choice and decision-making with respect to the specialty area and rural location. The study aims to shed light on the complex set of factors influencing career decision-making and to better understand the systems of influence on junior doctors. The study findings may contribute towards informing more effective ways to provide tailored career advice and develop career and training pathways to junior doctors and support those intending to pursue a rural career.

## 2. Materials and Methods 

This was a qualitative study informed by the principles of phenomenology, insofar as it was interested in participants’ lived experiences. Ethics approval was obtained by the WA Country Health Service (WACHS) Human Research Ethics Committee (HREC) 1130 in November 2018.

Participants were recruited among junior doctors in postgraduate years (PGYs) 1 to 5 undergoing training in Western Australia (WA) who participated in an online survey exploring the factors influencing the decision to pursue rural work administered in October 2018 and September 2019. A link to the survey was distributed to all postgraduate medical staff of the three primary employer health services in WA and the WA Country Health Service (WACHS) through their medical education officer using staff directories. A follow-up email was distributed two weeks later. Survey respondents were asked to include their contact details if they wished to be contacted for a follow-up interview, and those who responded to a follow-up email were invited to participate. A purposive sampling approach was adopted to ensure a broad representation of experiences.

Data was collected through semi-structured telephone interviews with an average duration of 33 min. The interview schedule covered four broad topics (work since graduation; career intentions; training and support; and future career plans) and consisted of general open-ended questions and prompts to explore specific aspects within each topic. The semi-structured format ensured the consistent exploration of all topics across all interviews whilst allowing for the introduction of new themes.

The process of data collection and analysis was iterative. All interviews were recorded and transcribed, and the resulting transcripts were imported into NVivo 12 [15] and subjected to thematic analysis. Data analysis was mainly inductive and followed the four steps described by Green and colleagues: Immersion in the data; coding; creating categories; and identifying themes [16]. Transcripts were read and coded separately by the research team, and a list of codes was developed and refined as coding progressed [17]. The conceptual framework by Pfarrwaller and colleagues—specifically the outer part of the model [12]—was used to guide the interpretation of the data, and the research team agreed on the main themes. Rigour was enhanced through investigator triangulation through team member checking, coding validation, peer debriefing [18], and the use of Nvivo [19].

## 3. Results

### 3.1. Sample

A total of 21 junior doctors were interviewed, and their characteristics are shown in Table 1.

### 3.2. Themes

Two main themes relating to the systems of influence on career decision-making emerged: (1) The importance of place and people, and (2) the broader context. These themes are discussed below and illustrated with contextualised quotes. Sites have been omitted to maintain confidentiality.

Place and people played a salient role in junior doctors’ explanations of their career decision-making. Work environment and geographical location featured prominently in participants’ narratives of their career trajectories and future career intentions, and their descriptions of these settings were inextricably linked to the people who inhabited them.

### 3.3. The Place: Lifestyle and Work Environment

Lifestyle factors were often cited by participants when discussing their past and future career choices, and these were associated with a specific ‘place’, either urban or rural. Junior doctors often mentioned social and recreational activities (spending time with family and friends, being part of sports and social clubs, and engaging in outdoor activities, such as camping or fishing), which they perceived as an important part of their wellbeing.

Although references to lifestyle factors were common among most participants, they were especially salient in the accounts of rural-based doctors. A rural location was associated with friendlier people, less stress, and a greater ability to maintain a good life–work mix, an aspect of professional practice which all participants, regardless of their location, valued highly. One doctor explained:
There’s aspects of the city I like—you know, like shopping and theatre and the various restaurants and ubering—but overall I don’t like traffic and I like a smaller community where people are friendly. So that was more appealing.(I13; Female; PGY5)

Similarly, another reflected:
I don’t think I could live in a big city now. As well as you can get from one side of town to the other in five minutes when there’s no traffic. You’ve got everything that you need in terms of shops. I’m not much of a city person, so yeah, just the lifestyle, being outdoors, camping and things like that.(I09; Female; PG3)

A peaceful, quiet natural environment was highly valued by all those with a strong rural intention; however, most expressed a preference for a town that is ‘small, but not too small’, which they described as providing a sense of community whilst still allowing them to maintain a certain degree of anonymity. This sense of community was strongly associated with a ‘country lifestyle’, even for those with a strong urban intention, as this comment by a doctor who earlier in the interview had expressed negative views about rural practice reveals:
I suppose one good thing about a rural community is that it does feel like home. It does feel like your whole world is in that small community. Because you know everyone. Because you’re aware of how things work. And it’s just very nice, but it can also be very isolating.(I01; Female; PGY3)

Urban-based doctors also spoke of the perceived lifestyle advantages of larger urban settings, such as the opportunity to engage in hobbies and financial pursuits and the ability to maintain their cultural and religious identity.

Another aspect of ‘place’ that featured prominently in the accounts of all participants was the importance of the work environment. Regardless of their practice location and career intentions, junior doctors highly valued characteristics, such as supportive and friendly teams, approachable senior staff, and easy access to consultants. Smaller teams were commonly perceived as enhancing the training experience, insofar as they allowed for personalised training and more opportunities to get hands-on experience and work alongside consultants.

Although not unique to the rural setting, positive characteristics of the work environment featured strongly in the accounts of those who had been exposed to rural training. Furthermore, these doctors tended to describe their experience by contrasting it to that to which they had been exposed to in an urban setting, as this quote illustrates:
I think the reason I get so much out of it is probably because everyone’s so supportive and encouraging up here. They will let you do a whole case and see a whole patient and do everything to do with that patient on your own, while still watching you and supporting you and making sure that you’re safe. I think that’s probably the main thing. You never feel like you’re being judged or being told off or anything. It’s really supportive. I feel like they’re really, really keen to teach but not in the same way as I experienced in the city hospitals where it’s just constant drilling with questions. It’s more of a friendly chat and advice on how to do things differently rather than ‘you’re doing this wrong’.(I09; Female; PGY3)

Similarly, another doctor explains:
I’ve been very impressed with the [rural hospital] ED senior staff. They’re amazing. I feel very well supported here and you never feel panicked or that you’re by yourself or that you can’t ask for help. So it’s a lot less stressful work environment than in the city. Because the [urban hospital] ED just didn’t feel the same, they weren’t approachable at all.(I08; Male; PG3)

This perceived contrast between the urban and rural training environment influenced attitudes towards rural practice broadly, and it also directly influenced career choices and intention, as one doctor who did his internship in an urban hospital reveals:
I opted to go to [rural location] under the surgical team for my surgical term and really, really enjoyed the smaller hospital, the more intimate and close-knit working environment and the smaller teams and how they kind of worked and do well together at first-name basis. When I went back to Perth to this big tertiary hospital with a whole lot of people who don’t know your name, it’s not as personable so to speak. I really missed and longed for that small work group that I experienced in [rural location].(I06; Male; PG2)

Although the collegiality and ease of access to senior staff that rural hospitals afforded was valued by most of those who had been exposed to rural training, a few commented that the lack of access to non-General Practitioner specialists in specific locations had at times detracted from their training. Overall, those who reported less positive experiences of their rural training environment tended to be urban-based doctors with no rural intention.

It was generally accepted that larger city hospitals were better equipped with respect to clinical infrastructure and diagnostic facilities; however, most doctors thought that the resources available at smaller hospitals were adequate to meet their training needs, and some even spoke of the benefit to their clinical reasoning of having limited access to diagnostic technology, where having to ‘make do’ with minimal equipment resulted in their becoming more independent thinkers.

### 3.4. The People: A Sense of Connectedness

Doctors in our study often referred to personal relationships when they discussed their career choices, and their narratives showed that the physical settings (workplace and geographical location) were intrinsically linked to the people inhabiting them. In this context, connectedness emerged as a factor influencing career decision-making, as relationships with colleagues, friends, family, and partners featured prominently in junior doctors’ attitudes towards and experiences of workplace and geographical location.

Positive interpersonal interactions with peers and senior staff were seen as enablers of positive practice by all junior doctors regardless of their specialty and rural/urban intention, as was being ‘familiar’ with the people around them and developing a sense of community. One doctor based at a large urban hospital explained that she had chosen the hospital because it was the most familiar to her, later adding:
It feels like a small community even though we’re in a tertiary hospital in the sense that faces are familiar.(I01; Female; PG3)

Similarly, other doctors reported having chosen their current workplace location based on the desire to maintain already established personal relationships, as this urban-based doctor explains:
I was very familiar with the place. I actually knew a few people working there already. It seemed like a nice place to work and I quite enjoyed being at [urban hospital], so I thought I might as well just stick around for a bit longer while working as well.(I16; Male; PGY3)

There was also evidence that the importance of relationships with peers extended beyond the workplace, especially for rural-based doctors, who valued developing friendships and socialising with their colleagues. Peers were also widely perceived as a valued source of support, and they also influenced career decision-making insofar as they were often cited as the main source of information on training and work opportunities.

Relationships outside of work (including family, friends, and partners) were also a strong influence on workplace location choices, and a desire to be in the same location or close to a partner was often cited as the main driver. A junior doctor who studied interstate reflects:
The current location was pretty much chosen solely because my partner is in Perth and I’ve spent four years long distance, so I thought it was time to come back and spend time with him. And also, my parents are here, and a lot of my family are here who I’m quite close to.(I07; Female; PGY2)

Similarly, another doctor explains how she chose the location for her rural generalist training program:
I obviously needed to and wanted to go rural. But for six months my partner couldn’t really change his job. So, it needed to be somewhere that was feasible, that wasn’t too far from Perth. […] It would have been very expensive in flights for us. Also, my parents are in [rural town], so it was nice to be closer to them for once in my last 10 years.(I13; Female; PGY5)

Personal factors were not always strong influences on career decision-making. Those whose narratives did not demonstrate that personal factors had influenced their choices tended to be urban-based doctors with a strong non-GP specialist intention along with a perception that this training was only available at an urban hospital. 

There was also evidence that personal relationships strongly influenced future career intentions and choices. Partners’ career progression was seen as one of the most important factors to take into consideration, featuring strongly even in the narratives of those who did not currently have a partner. In this context, an urban location was generally perceived as being more accommodating of the demands of two careers, and there was evidence that this had resulted in some junior doctors with a rural background and rural intentions opting for an urban-based career, as this quote reveals:
When I was in medical school, I was definitely thinking about going rural. That’s kind of changed a little bit. I think just having been in Perth for a really long time now, and my partner, she’s a lawyer so it’d be a bit difficult in terms of us both getting long-term work there. I think that’s swayed it to me wanting to stay metro more recently.(I14; Male; PG3)

Overall, accommodating partners’ careers (especially non-medical careers) was perceived as one of the main barriers to attracting and retaining doctors in the regions; this was seen as somewhat of a conundrum, and some reflected on the futility of trying to attract doctors to the regions if their partners could not secure employment.

In the broader context, career decision-making (regarding specialty and practice location) was also influenced by factors operating at a broader level. Perceptions of training and work opportunities, and training requirements strongly influenced career choices, and these were moderated by the information received. Other health system factors, including contractual arrangements and the sustainability of certain specialties, somewhat influenced career choices, as did perceptions of overarching policy and societal trends, although to a lesser extent.

### 3.5. Perceptions of Training and Work Opportunities

Perceptions of training and work opportunities strongly influenced career choices among all junior doctors, independently of their specialty and practice location intention. This was especially salient for vocational training choices but perceptions of training opportunities and gaps also influenced pre-vocational training choices. 

With regards to vocational training, a strong perception existed among most doctors that the only the rural training program available was a ‘rural generalist pathway’, based on primary care general practice (GP) with an extended skill set adapted to the demands of rural practice. This perception generated frustration among those with a strong rural intention and a non-GP specialist intention, who either appeared to resign themselves to enjoying what one doctor referred to as the ‘period of freedom’ in the country before relocating to the city for their vocational training, or chose to remain in the city. This doctor explains:
I actually find it very disappointing after working in rural areas and wanting to go back to those areas so badly, that unless you specifically want to be that rural GP, there’s firstly no pathway. And two, it’s not only not encouraged, it’s almost frowned upon. I find it amazing because the whole time I was in rural areas people talk about how much they’re trying to bring people rurally. When I look at it I kind of see a lot of closed doors.(I19; Male; FGY1)

Furthermore, there was some evidence that for those with strong rural intentions and still undecided about their specialty, perceptions of the availability of rural pathways influenced specialty preference. A rurally based doctor with a rural background explains:
To a certain extent, my choices—sort of anaesthetic, general surgery, GP surgery type of thing—are very much based on the idea that if I went and tried to do ENT surgery, I’d never really be able to be in the country. And I’m not a big fan of GP work because I don’t like to sit down for most of my day. But part of what I’m trying to do is find a specialty that would allow me to live in the country.(I04; Male; PGY3)

Similarly, another doctor reflected on his intention to apply for the GP training program under the Australian College of Remote and Rural Medicine, which operates under the ‘rural generalist pathway’ model:
I’m leaning towards that because I think ultimately no matter what specialty I go into I want to do something out in the country so that’s probably the best option for me at the moment.(I06; Male; PG2)

Early exposure to rural practice was seen as an effective way of attracting doctors to the regions but expanding rural training pathways was perceived to be critical to retaining them. At the time of the interview, a psychiatry rural training program existed at one regional location in WA. Two study participants held trainee psychiatry registrar positions at that location, and both strongly valued the opportunity the program afforded to pursue a rural career. Expanding the availability of rural training pathways was broadly seen as the most effective way of attracting and retaining doctors in the regions, and junior doctors with strong rural intentions who were aware of the existence of this program wished that a similar pathway existed for their chosen specialties. The following comment by a doctor who was planning to unwillingly relocate to Perth to apply for surgery training reflects this sentiment:
I think [psychiatry rural training] is a very, very big step in the right direction because I think you’ll find that those people will stay in the rural regions, and I think that that’s the way to foster it.(I12; Female; PGY5)

Similarly, another doctor commented:
I think the biggest thing for making people who want to live rurally stay is giving them the opportunity to do their specialty training programs in a rural area. Because the longer you stay somewhere, the more likely you’ll stay. But it doesn’t really help for people to stay in a country area for internship and residency if they can’t be a registrar and stay there long term.(I11; Female; PGY2)

There was also evidence that perceptions of pre-vocational opportunities influenced junior doctors’ practice location choices. There was a strong perception among some participants that rural doctors ‘miss out’ in terms of case exposure compared with their urban counterparts, and that this hinders their chances of being accepted into a vocational training program. This perception was reinforced by advice reportedly received from non-GP specialist consultants and resulted in some junior doctors with strong rural intentions making the decision to remain in an urban location during their pre-vocational years, and rural-based doctors planning to relocate to the city ahead of starting to apply for their preferred training program.

Rural-based early career doctors were keen to counter this perception, as one commented:
I guess talking to some of my colleagues, there is a perception that the actual patient load and the clinical—the variable presentations that you get isn’t as wide in the country as opposed to a metropolitan setting. […] And I think you can make an argument that in fact you get more attention at rural and regional settings.(I10; Male; PGY4)

Similarly, other rural-based doctors spoke of overcoming perceived training gaps by moving between rural sites to increase their exposure to a wider range of health profiles. Using technology was also seen as a way to address training gaps in the regions; simulation (SIM) training available at one rural site was especially highlighted, and being able to remotely access training sessions at other sites was seen as not only enhancing the teaching experience but also facilitating networking opportunities, especially with visiting non-GP specialists. One doctor who has hoping to be accepted into a rural training program commented:
I think the better access to training that we can get without having to relocate, without having to keep coming to metropolitan areas to get that. Technology helps with that, if we can have more availability of videoconferencing or—just improvement on what we’ve got at the moment I think is really, really necessary because it is one thing that really detracts from the area.(I03; Female, PGY2)

Perceptions of training and work opportunities were moderated by the information available and accessed by junior doctors. With regards to sources of information, career decisions tended to be based on information and advice received from colleagues and personal networks. There was evidence of information gaps regarding career pathways and training opportunities. Four participants admitted to still being undecided about their future career, and a further two were leaning towards community medicine were still unsure. One doctor explained:
I’m torn between anaesthetic training, general surgery training and possibly even GP surgery training out of—as Australian College of Rural and Remote Medicine (ACRRM). But I keep trying to research the GP surgery training part of things and it’s very vague and very, sort of, not clear as to how you go about doing it, and I’ve been trying to flesh that out if I can so I can understand exactly what’s entailed.(I04; Male; PG3)

Doctors with rural intentions wished for easier access to information on rural training opportunities, as there was a perception that the information was ‘out there’ but was not easy to find. Doctors also called for career advisors and rural mentors, who were seen as providing a valuable link to rural practice, especially for urban-based doctors with strong rural intentions. One urban-based doctor with a rural background and strong rural intention but still unsure about their specialty reflected:
I think mentors are really important in terms of your career, you have this idea of this role model but when you have been up studying now for your six years and then you work in a city environment and you do lose the ties or the mentors from rural doctors or rural GPs.(I17; Male; PGY3)

There were also information gaps relating to the bonded medical places (BMPs) scheme among the eight bonded doctors in our sample. The BMP data subset showed a spectrum of experiences relating to return-of-service planning obligations (RoSOs), ranging from those who reported having ‘maxed out’ on the pre-vocational and vocational training they could have recognised to those who were uncertain as to whether they would be able to, and those who did not intend to, fulfil their BMP obligations. Career planning—which included intentional urban terms to prepare for future rural practice, strong rural intention, and rural exposure (both undergraduate and prevocational)—appeared to positively influence RoSO planning. However, there were information gaps relating to the BMP scheme and RoSO even among those with a strong rural intention and clear career plans. 

### 3.6. Health System Factors

Junior doctors’ narratives of their career trajectories and intentions showed that health system factors also influenced career decision-making. Perceived administrative assistance, supportive work practices, and flexibility of contractual arrangements positively influenced doctors’ experience of their workplace, and they tended to be a stronger influence on career decision-making for those with a rural intention. The lack of unaccredited service registrar positions at regional hospitals was perceived as a barrier to career progression as many junior doctors thought that having service registrar experience would improve their chances of being accepted into increasingly competitive training programs.

Uncertainty around the future of the rural GP anaesthetic model appeared to be of concern to all four doctors who had been accepted into rural generalist programs at the time of the interview. They spoke of ‘tensions’ between GP anaesthetics training and specialist anaesthetics, and were concerned this would hinder their future hospital work, which all of them valued highly and wished to maintain. One doctor explained:
I’ve deliberately ruled out doing GP anaesthetics or GP obstetrics because I’m not sure of the future viability of those in the areas where I want to live and work. So I think just trying to read what’s going to be filled by all the huge numbers of doctors coming through that are going to fill specialist positions and then where the role of the GP is actually going and what we still be required to do to fill and where.(I08; Female; PGY3)

When asked about the factors that would be most important in her future career choices, another doctor reflected:
So finding a hospital that will still let GPs do anaesthetics because that’s getting tighter and tighter and more difficult. And finding a place that is going to have meaningful employment and a decent lifestyle for both my partner and I.(Int13; Female; PGY5)

Reflecting on her current practice location, the same doctor adds:
I like the location and the mix of work and it’s close to my family and it’s not too far from Perth. Because obviously a lot of my friends and contacts are still in Perth. So that is a nice sort of baby step position. […] but ultimately I think we will try and maybe consider going somewhere up in the Snowy Mountains because, you know, with the Snowy Hydro Scheme, there’s work for some engineers and that has a bit of a mix of GP based hospitals, as does Tasmania […] So potentially we will try the Eastern States if we don’t find a good mix for us in WA. And then potentially the other option is Canada. So Canada uses GP anaesthetists and they have a lot of mining. So that’s another possibility. I don’t know if we’d stay long-term in Canada because obviously, I’m a bit of a homebody and I love my family. But it might be a good place to try for 12 months to two years just for a bit of an adventure.(Int13; Female; PGY5)

As she reflects on her future career, this doctor provides an insight into the complex web of factors influencing career choices and operating at different levels: From personal relationships, accommodating the competing demands of two careers, and health system factors, to future population and workforce trends, both in Australia and overseas. 

## 4. Discussion

Our results demonstrate that medical career decision-making is a dynamic process that is influenced by a complex web of what Bronfenbrunner termed ‘ecological factors’ [14], and which Pfarwaller and colleagues incorporate as the ‘outer part’ of their model [12]. These factors operate at different levels, from the proximal (microsystem and mesosystem) to the distal (exosystem and macrosystem).

Factors operating at the micro- and mesosystem level strongly influenced career decision-making in our sample. Our data showed that junior doctors are part of four main settings (microsystems): Work environment, practice location, peer groups, and partners and family. Place and people were inextricably linked in the accounts of our junior doctors and played an important role in decision-making, demonstrating the importance of the interactions between components of the microsystem (the ‘mesosystem’ in Pfarwaller and colleagues’ model [12]).

Our findings on lifestyle as a career choice driver are consistent with other research [20,21], as is our data on the positive impact of supportive training teams [22,23]. As Australia continues to face the challenge of addressing health workforce shortages in rural areas, our data suggests that ‘selling’ the rural training environment, including positive characteristics of both place and people, would be an effective strategy to attract junior doctors to the regions. This strategy might prove particularly effective for ‘convertible’ [24] doctors who report enjoying a rural lifestyle and practice and disliking city living. Our data also suggests that a strong desire for a rural lifestyle might result in some junior doctors leaning towards a GP rather than a non-GP specialty on the basis of their perception that general practice is the only rural training program available. This has potential implications for workforce sustainability if doctors are opting for a career that is ultimately not aligned with their personal attributes and skill sets.

Our findings on the high value placed by junior doctors on the sense of connectedness to place and people are consistent with findings on the importance of practice and community among rurally trained GPs [21]. Similar to other research, our data showed that junior doctors’ career decision-making is influenced by personal and family reasons [11,23]. In contrast with research by Laurence and Elliot, who did not identify the role of partners’ careers in practice location choice [20], in our study, the career needs of partners (and, interestingly, of potential partners) strongly influenced career choices and long-term career planning, adding contextual evidence to the survey data reported by Nichols and colleagues [23]. 

A number of factors operated at the ‘exosystem’ [12,14] level. Perceptions about training and work opportunities influenced career choices, and these perceptions were moderated by the information received. Our findings on the role of perceptions of a lack of rural training opportunities on career choice are consistent with other research [11,20,23], especially with the data on RCS graduates reported by Eley and colleagues [11] and research conducted with junior doctors exposed to a rural community practice training term [23].Our data showed that for some junior doctors with a strong rural intention and a non-GP specialty preference, a perceived lack of rural training pathways generated frustration, somewhat lending support to the ‘four quadrant model’ described by Stagg and colleagues [24], although in our study a rural background did not seem to be associated with the profile of ‘frustrated’ [24].

Our findings regarding the importance of ‘word of mouth’ as a source of information and its impact on decision-making are consistent with other research conducted with recent GP graduates [21]. Limited data from our study suggesting that professional advice reportedly received by junior doctors might contribute to misperceptions on rural career opportunities warrants further investigation. These findings, combined with our evidence on information gaps regarding career pathways generally and return-of-service obligations under the BMP scheme, lend support to calls for the need to develop a comprehensive multi-channel information provision strategy [25]. 

Our findings on the impact of health system factors, including perceived administrative assistance, supportive work practices, flexibility of contractual arrangements, lack of unaccredited registrar positions and the sustainability of certain specialties, are underreported and warrant further exploration. Also, further research is needed to understand how these factors change over time.

Our study has strengths and limitations. We included a broad range of junior doctors regardless of practice location, specialty intention, and rural exposure, and we successfully tested one component of Pfarrwaller and colleagues’ model [12] to demonstrate how different systems of influence on career decision-making operate. Furthermore, we explored career decision-making ‘in real time’ rather than retrospectively. We acknowledge some limitations. Firstly, participants self-selected and, given our recruitment strategy, our sample included an overrepresentation of junior doctors with an interest in rural practice; this, however, does not preclude the validity of our results. Secondly, our study focused on one state with a distinctive population distribution and geographic characteristics, and as a result, some of our findings might have limited applicability in other settings.

## 5. Conclusions

In conclusion, we found that career decision-making among junior doctors is influenced by a complex web of factors operating at different levels. Our findings demonstrate that purposefully, rural-branded postgraduate training pathways, especially for non-GP specialists, and improved career information delivery flow are urgently needed to address rural workforce shortages. Our findings also show that attraction and retention strategies must address personal and family factors if they are to be successful, and that promoting the rural lifestyle and work environment, and increasing the resources allocated to assisting partners find work might prove effective. 

Government and postgraduate college rural training initiatives must be well promoted and branded, demonstrating integrated training pathway options that engage junior doctors and their peer networks. As Australia faces the challenge of developing a sustainable rural health workforce, developing innovative, flexible strategies that are responsive to the individual aspirations of its workforce whilst still meeting its healthcare service delivery needs will provide a way forward.

## Figures and Tables

**Table 1 ijerph-17-00537-t001:** Interview participants’ main sample characteristics.

Characteristic			Total
Sex	Female		11
	Male		10
		Total	21
PGY (at interview)	PGY1		*4*
	PGY2		4
	PGY3		8
	PGY4		3
	PGY5		2
		Total	21
Rural Background	Yes		11
	No		10
		Total	21
RCS Participation	Yes		14
	No		7
		Total	21
Specialist Intention	GP		7
	Non-GP		8
	Unsure		6
		Total	21
Age Group	25–30		17
	31–40		4
		Total	21

GP: General practitioner.

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
