# Peer review of "Understanding the Factors Influencing Junior Doctors’ Career Decision-Making to Address Rural Workforce Issues: Testing a Conceptual Framework"

_ijerph, 2020, doi:10.3390/ijerph17020537_

Round 1

Reviewer 1 Report

The subject of the article is interesting and it is linked to the objectives of the journal, however, there are a number of issues that have to be reconsidered.

For a better visibility on databases, the authors are asked not to repeat among keyword the words/concepts included on the title of the article.

The Conclusion part is rather superficial, based on the quantity of data, it can be improved to offer more practical solutions.

Author Response

Thank you for the reviewer comments and suggestions which have been incorporated into the resubmitted manuscript (tracked changes).

Point 1

For a better visibility on databases, the authors are asked not to repeat among keyword the words/concepts included on the title of the article.

Response Point 1

The keywords have been changed to improve database visibility by using MeSH descriptors.

Point 2

The Conclusion part is rather superficial, based on the quantity of data, it can be improved to offer more practical solutions.

Response Point 2

Additional practical conclusions based on further elaboration in the discussion are included in the review.

Reviewer 2 Report

The article provides relevant evidence on the factors explored, through a rigorous and consistent methodology.

Some comments are suggested:

The use of MeSH descriptors is recommended as keywords. Although the introduction highlights the importance of considering the factors that influence the decision-making capacity of newly graduated physicians, it requires a greater explanation of these factors. It would be interesting to describe the studies that have been raising what elements influence and how these have impacted on decision-making. The last paragraph of the introduction should only clearly and concisely describe the objective of the study but not what type of study has been developed. The authorization number of the WACHS Human Research Ethics Committee (HREC) must be provided. Table 1 of the results could be improved in terms of distribution to make it more reader friendly. In the methodology it should be explained that methodological rigor procedures have been followed as would be triangulation.

Author Response

Thank you for the comments and suggestions which have been included in the reviewed submission (Tracked changes).

Point 1

The use of MeSH descriptors is recommended as keywords.

Response Point 1

MeSH descriptors are used for new keywords.

Point 2

Although the introduction highlights the importance of considering the factors that influence the decision-making capacity of newly graduated physicians, it requires a greater explanation of these factors. It would be interesting to describe the studies that have been raising what elements influence and how these have impacted on decision-making. The last paragraph of the introduction should only clearly and concisely describe the objective of the study but not what type of study has been developed.

Response Point 2

Further explanatory components have been added to the introduction to describe the theory of decision making (Pfarwaller et al) tested in the paper with the last para of the introduction altered to be more concise.

Point 3

The authorization number of the WACHS Human Research Ethics Committee (HREC) must be provided.

Response Point 3

The WACHS HREC number has been included.

Point 4

Table 1 of the results could be improved in terms of distribution to make it more reader friendly. 

Response Point 4

Table 1 has remained the same as alternatives tried (and tested) did not appear to be more user friendly.

Point 5

In the methodology it should be explained that methodological rigor procedures have been followed as would be triangulation. 

Response Point 5

The methodology section has been strengthened to demonstrate the extent of rigour in the analysis.

Reviewer 3 Report

Sample size is too small to arrive at any conclusions, The parameters need to be more objectively defined. The main parameters and sub-parameters are not statistically described and thus the whole is paper is more subjective. There is no statistical basis for any conclusions.

Author Response

Thank you to the reviewer for the comments.

Point 1

Sample size is too small to arrive at any conclusions, The parameters need to be more objectively defined. The main parameters and sub-parameters are not statistically described and thus the whole is paper is more subjective. There is no statistical basis for any conclusions. 

Response to Point 1

It appears that the reviewer may be unfamiliar with the qualitative nature of the study and therefore the methodology and results. The sample is small due to experiencing data saturation. No power calculations are required to ensure statistical significance as is required in quantitative studies.

Round 2

Reviewer 3 Report

1.Limitations of the study should have been discussed .

2. Sample size should be bigger.